# Long-Term Outcomes among Patients with Prolonged Disorders of Consciousness

**DOI:** 10.3390/brainsci13020194

**Published:** 2023-01-23

**Authors:** Yan Liu, Xiao-Gang Kang, Qiong Gao, Yu Liu, Chang-Geng Song, Xiao-Jing Shi, Jia-Ning Wu, Wen Jiang

**Affiliations:** Department of Neurology, Xijing Hospital, Fourth Military Medical University, Xi’an 710032, China

**Keywords:** disorders of consciousness, unresponsive wakefulness syndrome, minimal consciousness state, mortality, prognostic

## Abstract

Purpose: To evaluate the long-term survival and functional outcomes of patients with prolonged disorders of consciousness (pDoC) 1–8 years after brain injuries. Methods: Retrospective study to assess the long-term survival and functional outcomes of patients with pDoC was conducted. We performed Cox regression and multivariate logistic regression to calculate hazard ratios (HRs) for the outcome of survival and to identify risk factors of the functional outcome. Results: We recruited 154 patients with pDoC. The duration of follow-up from disease onset was 1–8 years. The median age was 46 years (IQR, 32–59), and 65.6% (*n* = 101) of them were men. During the follow-up period, one hundred and ten patients (71.4%) survived; among them, 52 patients had a good outcome. From the overall survival curve, the 1-, 3-, and 8-year survival rates of patients were about 80.5%, 72.0%, and 69.7%, respectively. Cox regression analysis revealed a significant association between the lower APACHE II score (*p* = 0.005) (cut-off score ≥ 18) and the presence of sleep spindles (*p* = 0.001) with survival. Logistic regression analysis demonstrated a higher CRS-R score (cut-off score ≥ 7), and presence of sleep spindles were related to a favorable outcome among patients with pDoC. Conclusions: Sleep spindles are correlated with both long-term survival and long-term functional outcome in pDoC patients.

## 1. Introduction

After the first days of coma resulting from critical brain injury, some patients may evolve to severe disorders of consciousness (DoC), remaining in a vegetative state/unresponsive wakefulness syndrome (VS/UWS) or minimally conscious state (MCS). Many of these patients can survive for a long time under the care of families or medical institutions, which persists for more than 28 days, known as prolonged DoC (pDoC) [1,2,3]. These severe clinical conditions might evolve towards the recovery of full consciousness or persist unchanged for years, or they might worsen until death. An accurate and reliable assessment of the outcomes of patients with DoC is of great importance for their management and care. Most studies typically concentrate on the first 12 months after onset, while little is known about the long-term outcome of pDoC patients after 12 months.

There are few long-term prognostic studies that have investigated pDoC patients 1–16 years after brain injury [4,5,6,7,8,9,10,11]. These studies indicated that recovery might be possible after more than 12 months, and some patients continue to improve for several years post-injury. The most common predictors are traumatic etiology, younger age, diagnosis of MCS compared to VS/UWS, and shorter time since injury associated with favorable outcomes. Moreover, the absence of median nerve somatosensory evoked potentials, worse electroencephalographic (EEG) background activity, and a lower CRS-R score (<6 points) were also considered predictors for mortality and poor outcome [12,13]. However, identifying a patient’s routine clinical variables to predict the potential of long-term recovery is scarce for pDoC patients.

In this longitudinal observational study, we describe a large group of VS/UWS and MCS patients with various etiologies 1–8 years after brain injuries. The objectives of the current study were to evaluate the long-term survival and functional outcomes of patients with pDoC and to analyze the risk factors affecting their long-term prognosis with readily available routine clinical variables.

## 2. Materials and Methods

### 2.1. Study Setting and Participants

This retrospective study was performed based on a registry for pDoC patients from the neurological intensive care unit (N-ICU) at Xijing Hospital, one of the largest tertiary academic hospitals in northwest China. All patients were recruited for this study from January 2014 to December 2020. The inclusion criteria were: (1) age 16–80 years; (2) diagnosis of VS/UWS or MCS according to the Coma Recovery Scale Revised (CRS-R) score within the first week after admission (CRS-R was assessed 5 times within the first week after admission, and the best score was used as the basis for diagnosis); (3) time post-onset > 28 days. Exclusion criteria were: (1) previous history of acquired brain injury or psychiatric or neurodegenerative diseases; (2) coexisting neoplasms, severe organ dysfunction, or unstable clinical condition (e.g., hemodynamic instability or severe respiratory failure); (4) time post-onset > 3 months; (4) patients who lost to follow-up. The study was approved by the ethics committee of the Xijing Hospital and was carried out in agreement with Chinese laws and the Helsinki declaration.

### 2.2. Data Acquisition

Demographic, clinical, and neurophysiological information related to the recovery of patients with pDoC were collected. The following demographic data were recorded: (1) age; (2) sex; (3) residence (urban or rural); and (4) educational level (≤high school diploma or >high school diploma) [14]. Clinical variables included: (1) etiology; (2) clinical diagnosis; (3) CRS-R score [15]; (4) Acute Physiologic and Chronic Health Evaluation (APACHE) II score [16]; (5) length of stay in NICU; (6) time post-injury; (7) Charlson Comorbidity Index (CCI), classified into three categories: 0 point, 1 to 2 points, 3 points [17,18]; (8) motor responses to painful stimulation, coded as flexion withdrawal of at least one limb, or absent/extensor motor responses [12]; and (9) serum neuron-specific enolase(NSE) level dichotomized as <33 ng/mL or ≥33 ng/mL [19]. Three variables concerned neurophysiological parameters: (1) sleep spindle, a kind of waveform distinct from the background with a frequency between 12 and 16 Hz, duration between 0.5 and 2 s and occurring in the context of EEG activity; (2) EEG reactivity, defined as a change in the frequency or amplitude of the background activity with a precise time-locked correlation to external stimulation [12]. Three stimuli were conducted in turn to assess EEG reactivity. These three stimuli included passively opening or closing the patient’s eyes (that is, the patient’s eyes was passively opened when his/her eyes were closed and passively closed when his/her eyes were open), calling patients by their name (auditory stimulation), and painful stimulation on the nail bed of the thumb. Each stimulus lasted 5 s and repeated three times with intervals of 60 s. The stimulation was carried out in the order from less noxious to more noxious. (3) Synek scale was coded as a Synek score ≥ 3 or <3 [20,21]. NSE and EEG-markers were collected within 3 days after admission, and other data were collected within the first week after admission.

### 2.3. Definition of Outcome

The primary outcome was the survival of pDoC patients. For the survival analysis, the time-to-event was calculated as the number of days from initial study enrollment to the date of survival until the end of the follow-up period. A structured telephone interview was conducted for 1–8 years post-injury by a trained neurologist who was blinded to the clinical data. The secondary outcome was the patient’s functional outcome assessed by the Glasgow Outcome Scale–Extended (GOS-E) [22], an eight-point scale ranging from 1 (death) to 8 (upper good recovery). For the purpose of statistical analysis, the GOS-E was dichotomized as a favorable neurologic outcome (GOS-E score 4–8) and an unfavorable outcome (GOS-E score 1–3) [23].

### 2.4. Statistical Analyses

The analyses were conducted using SPSS 22 (IBM SPSS Statistics for Windows, Version 22.0) and R software (Institute for Statistics and Mathematics, Vienna, Austria; version 3.2.2). Continuous variables were expressed as mean ± standard deviation (SD) or median (interquartile range, IQR), and categorical variables were expressed as numbers and percentages. The significant differences between different outcome groups were analyzed using a Student-*t* test or Mann-Whitney U test for continuous variables and an χ2 test for categorical variables. We performed Cox regression, reporting hazard ratio (HRs), and 95% confidence intervals (CIs) for the outcome of survival. Kaplan-Meier curves were conducted and tested for differences in the curves using the log-rank test. Censoring was performed at the time of death or 31 December 2021, the last date available in our dataset. A stepwise multivariate logistic regression was used to identify risk factors of the functional outcome. All tests were two-sided, and a *p* value of less than 0.05 was considered statistically significant in Cox and logistic analyses, as was a *p* value lower than 0.005 for comparison tests on survival curves for different causes.

## 3. Results

A total of 170 patients with UWS or MCS were recruited. Six patients were excluded from the study due to the previous history of acquired brain injury and four patients with postinjury > 3 months. Six patients were lost to follow up. A total of 154 patients were used for final analysis. The duration of the follow-up from disease onset was 1–8 years. Figure 1 shows a flow chart of the studied population. Demographic and baseline characteristics are summarized in Table 1. The median age was 46 years (IQR, 32–59), and 65.6% (*n* = 101) of them were men. There were 108 patients diagnosed with UWS, with a median CRS-R score of 7 points (IQR, 5–9). As for the place of residence, 107 patients lived in rural areas, while 47 lived in urban areas. Most patients only completed secondary education (*n* = 83, 53.9%). The median length of stay in the ICU was 18 days (IQR, 12–29), the median time postinjury was 46 days (IQR, 36–64), and the median APACHE II score was 13 points (IQR, 10–17). The main cause of pDoC in our sample was anoxic brain injury (ABI) (*n* = 55, 35.7%), followed by vascular lesions (VL) (i.e., hemorrhagic or ischemic) (*n* = 38, 24.7%), infectious diseases (*n* = 25, 16.2%), traumatic brain injuries (TBI) (*n* = 23, 14.9%), and other causes (*n* = 13, 8.4%). One hundred and ten patients (71.4%) survived; among them, 52 patients had a good outcome. Among the 102 patients (66.2%) classified as poor outcome, 44 deaths were included. No patients died because of withdrawal of life-sustaining treatment. Despite life-sustaining treatment, death was due to serious medical complications (e.g., sepsis, acute respiratory distress, gastrointestinal bleeding, cardiac arrest). Overall, the good recovery rate was 49.9% in MCS patients and 27% in UWS patients.

During an average 4.4 ±3.4-year follow-up period (the number of patients at each follow-up period were as follows: 20 (13.0%) patients were followed up for 1 year, 17 (11.0%) for 2 years, 19 (12.4%) for 3 years, 24 (15.6%) for 4 years, 23 (14.9%) for 5 years, 17 (11.0%) for 6 years, 16 (10.4%) for 7 years, and 18 (11.7%) for 8 years), a total of 44 (28.6%) patients died. From the overall survival curve, the 1-year survival rate of patients was about 80.5%, and the 3-year survival rate was approximately 72.0%. The following survival rate decreased slowly, with an 8-year survival rate of 69.7% (Figure 2A). There was no significant difference (*p* = 0.243) in long-term survival between VS/UWS patients and MCS patients. The 1-year survival rate of UWS patients was 78.6%, and that of MCS patients was 84.8%. The 8-year survival rate of UWS and MCS patients was 66.8% and 76.7%, respectively (Figure 2B).

In addition, patients with TBI (73.9%) had the highest long-term survival rate during the 8-year follow-up, followed by VL (72.1%), infectious diseases (68.0%), and ABI (65.0%). As shown in Figure 2C, there was a significant difference of survival between TBI and ABI in the log-rank test (*p* < 0.001). However, no significant difference was found between other etiologies. Fifty-two (33.8%) of the total patients presented a favorable outcome, while 102 (66.2%) had an unfavorable outcome at study termination. MCS patients achieved a higher probability in terms of functional independence than UWS patients (49.9% vs. 27.0%). Patients with TBI had the highest proportion of a favorable outcome (*n* = 11, 47.8%), followed by infection (*n* = 9, 36.0%), ABI (*n* = 18, 32.9%), other diseases (*n* = 4, 30.8%), and VL (*n* = 10, 26.3%) (Figure 3).

The univariate Cox proportional hazard model revealed that patients with a higher education level (*p* = 0.010), a lower APACHE II score (*p* = 0.001), a presence of sleep spindles (*p* < 0.001), and EEG reactivity (*p* = 0.005) correlated significantly with the outcome of survival. Multivariate Cox proportional hazard models confirmed a significant association between lower APACHE II score (*p* = 0.004) (cut-off score ≥ 18) and the presence of sleep spindles (*p* = 0.001) with survival (Table 2).

Univariate analysis indicated that clinical diagnosis (*p* = 0.006), CRS-R score (*p* = 0.001), presence of sleep spindles (*p* = 0.010), and EEG reactivity (*p* = 0.032) showed significant differences between patients who had good outcomes and those who had poor outcomes (Table 3). By using multivariate logistic regression analysis, a higher CRS-R score (cut-off score ≥ 7) and a presence of sleep spindles were related to an increased probability of favorable outcome among patients with pDoC.

## 4. Discussion

In this study, we explored the long-term clinical outcomes of VS/UWS and MCS patients, observed the clinical evolution of pDoC patients, screened out risk factors, and then established prediction models to investigate their impact on the long-term prognosis of pDoC patients. Our results indicated that APACHE II score and sleep spindles were associated with long-term survival. An APACHE II score ≥ 18 and the presence of sleep spindles made it more likely for pDoC patients to survive. CRS-R score and sleep spindles were independent predictors for functional outcome. A CRS-R score ≥ 7 and the presence of sleep spindles were related to a favorable outcome in pDoC patients.

A strength of this study was the finding that sleep spindles are correlated with both long-term survival and long-term functional outcome in pDoC patients. Previous studies indicated that the presence of sleep features and sleep-wake cycles seem to be related to the severity of consciousness impairment [24,25,26]. They correlate with higher residual functioning and can provide insight into covert recovery in the chronic DoC. Sleep organization may reflect both the integrity of consciousness-supporting brain networks and the engagement of those networks during wakefulness in pDoC. Specific sleep characteristics, such as the presence of sleep spindles identified as hallmarks of non-REM sleep, are an important marker for the prognosis of recovery from DoC [27]. The prognostic value of the sleep spindles for clinical outcomes was demonstrated in several studies on pDoC [12,28]. However, data on the association between absent sleep spindles and mortality in pDoC patients are scarce. The study demonstrated the important value of sleep spindles for long-term prognosis of patients with pDoC.

In clinical variables, the CRS-R score was significantly associated with a better outcome, which is in line with others studies [29,30,31,32]. However, it is not a significant predictor of mortality. This finding is in contrast with previous studies reporting an association of CRS-R score at study entry with mortality [13]. This may be due to their study comparing the CRS-R score ≥ 5 groups, whereas our study did not classify the CRS-R score. Previous studies have also shown that MCS patients showed higher functional improvement and lower mortality compared to UWS patients [4,6,7,9,33,34,35]. In our study, it is shown that a clinical diagnosis of UWS/VS or MCS at ICU admission had no significant association with outcome. Nevertheless, the likelihood of survival was significantly higher for MCS patients compared to VS/UWS patients, and there was a good recovery rate of 49.9% in MCS patients and of 27% in UWS patients. These results were similar with those from Steppacher et al. [7], but they were higher than those reported in other literature [6,9]. We considered that the reasons for these differences might be related to the differences in time to follow-up [4,36] and time from onset to study entry [37]. This may indicate that the initial state of consciousness is not a critical factor for the patient’s long-term outcome. This is consistent with the view of Estraneo et al. [9]. The severity of illness scoring systems, such as the APACHE II score, is used widely for the evaluation of disease severity and mortality prediction, especially for ICU patients [38]. Our study suggests that the APACHE II score is also an independent risk factor for mortality in pDoC patients.

The 8-year cumulative survival rate in pDoC patients was 69.7%, which was consistent with the results of Pascarella et al., whose study showed that the six-year survival rate hovered around 70% [39]. Previous studies have demonstrated that age is an important predictor of functional outcome [6,40,41], while no effect of age on long-term functional outcome was found in our study. Likewise, there was no association between age and long-term mortality in our study, which is inconsistent with a previous study [13]. This may be because the related indicators reflecting residual brain function, such as sleep spindles and some important clinical scales, are more valuable in a long-term prognosis. The study demonstrated that mortality after traumatic brain injury is associated with less education [14]. Univariate Cox regression analysis indicated that a lower educational level was an independent risk factor for predicting death, probably because these individuals had limited access to rehabilitation resources. Notably, patients who live in rural areas have limited access to medical resources, which may have an impact on long-term clinical outcomes, and larger sample sizes are needed to verify the result.

Medical comorbidities and their severity are critical determinants of patient clinical outcomes and should be considered in the prognostic variable. In this study, no impact of complications on long-term functional outcome and survival in pDoC patients was found, possibly due to clinical complications requiring appropriate management and timely treatment, which in turn reduced clinical and neurological complications and mortality. It has no predictive value for etiology and is inconsistent with other studies [42,43,44], possibly because the sample size is too small (*n* = 23, 15.1%). However, based on the KM curve and GOS-E score distribution, the prognosis of TBI patients was significantly better than that of patients with other etiologies. In a study of UWS patients, Kang et al. showed that motor response was shown to be an independent predictor of awareness recovery [12]. Our results suggest that motor response is not a predictor of long-term outcomes. Several studies found that short-term mortality and unfavorable outcome were significantly associated with greater NSE concentrations [12,45,46]. No statistical significance was found in our study, probably because NSE is mainly used to predict mortality and neurological outcome in traumatic brain injury patients.

There were some limitations in our study. First, this is a retrospective cohort study with a small sample size from one tertiary hospital, and prospective multicenter well-designed studies are required to verify our findings. Second, the variables included in this study are not exhaustive. Given the issues of expense and accessibility, several known and suggested variables such as fMRI and positron emission tomography (PET) were not included in the analysis. However, future studies should consider the role for these variables. Finally, the level and intensity of the care and rehabilitation program for pDoC patients may have differences, which in turn could affect the outcome of patients to some extent.

## 5. Conclusions

In conclusion, our results indicated that sleep structural elements combined with important clinical scales can provide valuable long-term prognostic information for patients with pDoC. The APACHE II score and sleep spindle-based model had predictive value for long-term survival, while the CRS-R score and sleep spindle-based model had predictive value for long-term functional outcome in pDoC patients. Future prospective studies are required to validate the findings.

## Figures and Tables

**Figure 1 brainsci-13-00194-f001:**
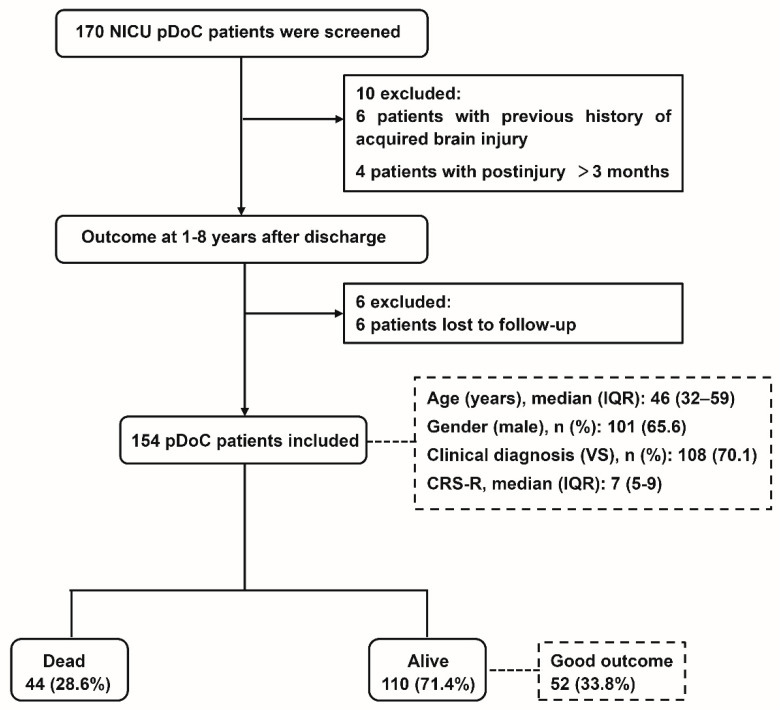
Flow diagram showing patient recruitment and outcomes.

**Figure 2 brainsci-13-00194-f002:**
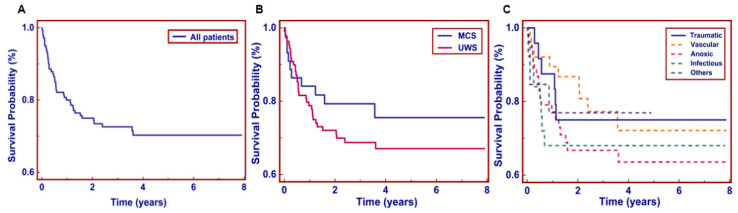
Kaplan-Meier plots for the survival rate in patients with prolonged disorders of consciousness. (**A**) Kaplan-Meier curves are shown for all 154 patients; (**B**) survival rates are shown depending on diagnosis; (**C**) survival rates based on different etiologies. MCS: minimally conscious state; UWS: unresponsive wakefulness syndrome.

**Figure 3 brainsci-13-00194-f003:**
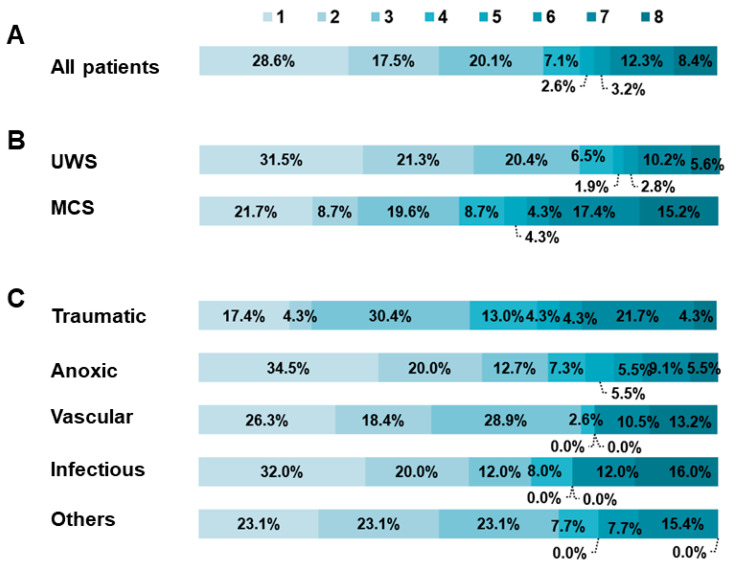
Functional outcomes according to the Glasgow Outcome Scale-Extended (GOS-E): (**A**) distribution of GOS-E scores to all patients; (**B**) distribution of GOS-E scores to different clinical diagnosis; (**C**) distribution of GOS-E scores to different etiologies; poor outcome: 1. dead, 2. vegetative state, 3. lower severe disability; good outcome: 4. upper severe disability, 5. lower moderate disability, 6. upper moderate disability, 7. lower good recovery, 8. upper good recovery. MCS: minimally conscious state; UWS: unresponsive wakefulness syndrome.

**Table 1 brainsci-13-00194-t001:** Demographic, clinical, and neurophysiologic data at study entry.

Variable	Total (*n* = 154)	Alive (*n* = 110)	Dead (*n* = 44)
Age, y	46 [32,59]	45 [31,55]	54 [33,63]
F/M	53/101	35/75	18/26
Residence, rural/urban	107/47	79/31	28/16
Education, y < 12 / ≥12	83/71	67/43	16/28
Etiology, T/NT	22/132	16/94	6/38
Length of stay in the ICU, d	18 [12,29]	17 [12,29]	21 [12,30]
TPI, d	46 [36,64]	45 [34,60]	48 [37,78]
CRS-R score	7 [5,9]	7 [5,9]	6 [5,8]
Clinical diagnosis, VS/MCS	108/46	74/36	34/10
APACHE II score	13 [10,17]	12 [10,15]	15 [11,18]
CCI, 0/1–2/ ≥3	55/66/33	38/50/22	17/16/11
Motor response, P/A	103/51	76/34	27/17
NSE (μg/L), <33/ ≥33	69/85	53/57	16/28
Sleep spindles, P/A	84/70	70/40	14/30
Synek’s scale, <3/ ≥3	90/64	67/43	23/21
EEG R, P/A	117/37	90/20	27/17

Abbreviations: F/M: female/male; T/NT: traumatic/not traumatic; MCS: minimally conscious state; VS/UWS: vegetative state/unresponsive wakefulness syndrome; ICU: intensive care unit; TPI: time post-injury; CRS-R: Coma Recovery Scale–Revised; APACHE II: Acute Physiologic and Chronic Health Evaluation II; CCI: Charlson Comorbidity Index; NSE: neuron specific enolase; EEG: electroencephalogram; EEG-R: EEG reactivity; P/A: present/absent; d: days; y: years.

**Table 2 brainsci-13-00194-t002:** Univariate and multivariate Cox proportional-hazard model for the risk of death of pDoC patients.

Variable	Univariate Analysis		Multivariate Analysis	
	HR (95%CI)	*p*	HR (95%CI)	*p*
Age	1.014 (0.995–1.033)	0.160		
Sex	0.755 (0.414–1.376)	0.358		
Residence	1.320 (0.713–2.441)	0.377		
Education	2.252 (1.218–4.165)	0.010		
Etiology	0.910 (0.385–2.154)	0.831		
Clinical diagnosis	0.659 (0.325–1.334)	0.246		
Length of stay in the ICU	1.008 (0.997–1.020)	0.168		
TPI, d	0.762(0.411–1.432)	0.193		
CRS-R score	0.958 (0.862–1.066)	0.435		
APACHE score	1.114 (1.045–1.188)	0.001	1.101 (1.031–1.177)	0.004
APACHE score ≥ 18	0.290 (0.153–0.550)	<0.001		
CCI, 0		0.559		
1–2	0.749 (0.378–1.484)	0.408		
≥3	1.106 (0.518–2.362)	0.794		
Motor response	0.797 (0.435–1.463)	0.465		
NSE (μg/L)	0.627 (0.339–1.160)	0.137		
Sleep spindles	0.314 (0.166–0.594)	<0.001	0.347 (0.183–0.659)	0.001
Synek’s scale	0.789 (0.436–1.425)	0.432		
EEG R	0.416 (0.227–0.764)	0.005		

Abbreviations: ICU: intensive care unit; TPI: time postinjury; CRS-R: Coma Recovery Scale–Revised; APACHE II: Acute Physiologic and Chronic Health Evaluation II; CCI: Charlson Comorbidity Index; NSE: neuron specific enolase; EEG: electroencephalogram; EEG-R: EEG reactivity; CI: confidence interval; HR: hazard ratio.

**Table 3 brainsci-13-00194-t003:** Univariate and multivariate logistic regression analyses.

Variable	Univariate Analysis		Multivariate Analysis	
	OR (95%CI)	*p*	OR (95%CI)	*p*
Age, y	0.980 (0.960–1.001)	0.061	0.978 (0.955–1.000)	0.051
Sex, M	1.281 (0.627–2.615)	0.497		
Residence, urban	0.770 (0.368–1.615)	0.489		
Education, y ≥ 12	1.127 (0.577–2.201)	0.726		
Etiology, T	2.220 (0.890–5.533)	0.087		
Clinical diagnosis, MCS	2.724 (1.329–5.584)	0.006		
Length of stay in the ICU, d	0.993 (0.976–1.009)	0.386		
TPI, d	1.114 (0.867–1.415)	0.114		
CRS-R score	1.245 (1.097–1.412)	0.001	1.258 (1.095–1.445)	0.001
CRS-R score ≥ 7	2.810 (1.390–5.680)	0.004		
CCI, 0		0.443		
1–2	1.000 (0.475–2.104)	1.000		
≥3	0.569 (0.213–1.473)	0.240		
APACHE score	0.938 (0.867–1.015)	0.112		
Motor response, *p*	1.345 (0.652–2.776)	0.422		
NSE (μg/L), <33	1.221 (0.624–2.386)	0.560		
Sleep spindles, *p*	2.531 (1.250–5.126)	0.010	2.368 (1.131–4.958)	0.022
Synek’s scale, <3	1.371 (0.690–2.724)	0.367		
EEG R, *p*	2.679 (1.086–6.609)	0.032		

Abbreviations: F/M: female/male; T/NT: traumatic/not traumatic MCS: minimally conscious state; ICU: intensive care unit; TPI: time postinjury; CRS-R: Coma Recovery Scale–Revised; APACHE II: Acute Physiologic and Chronic Health Evaluation II; CCI: Charlson Comorbidity Index; NSE: neuron specific enolase; EEG: electroencephalogram; EEG-R: EEG reactivity; P/A: present/absent; d: days; y: years; OR: Odds ratio; CI: confidence interval.

## Data Availability

The study data will be available from the corresponding author upon reasonable request.

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
