# Peer review of "Long-Term Outcomes among Patients with Prolonged Disorders of Consciousness"

_brainsci, 2023, doi:10.3390/brainsci13020194_

Round 1

Reviewer 1 Report

This is an interesting study focusing on long-term consequences (mortality and disability) after DOC. Relatively few natural history and prognostic studies have reported long-term functional outcomes on these population after one year, so prognosis after that date can be challenging. Recent clinical guidelines have emphasized the need for more clinical information focused on long-term care so results presented here are in line with this interest.

A lot of work has been involved in this study, yet I would like to raise few essential questions before final acceptance.

Methods

-       (line 60). Please specify the number and frequency of CRS-R needed for diagnosis at inclusion (if just one, it should be explained as a limitation).

-       “time post-onset >28d and < 3 months” (lines 61 and 64) . Please provide length of time postinjury, specifically chronicity at the moment of the recruitment. It is not clear along the document whether some of the variables were assessed at ICU admission (line 224) or at enrollment, and this is highly relevant in terms of the characteristic of the sample and for comparison with previous studies (DOC versus pDOC)

-       “Duration of follow-up was 1-8 years” (line 89) and “average 4.4+-3.4 year follow-up period” (line 140). It would be helpful to know the number of number of patients at each follow-up period or at least the number/% of patients followed up to 8 years after injury.

-       Data acquisition (line 68-84). Please specify the time of acquisition (within one week after admission??, after recruitment,??...)

-       “EEG reactivity” (line 81) seems to be a very relevant variable according to the results of the regression models. The authors include a bibliographic reference but it would be helpful if they can define the protocol and the scoring procedure more accurately. 

-       “Favorable neurologic outcome (GOS-E score 4-8) and unfavorable outcome (GOS-E score 1-3) (line 93-94)”. Please justify (Faugueras F et al. Brain Inj. 2018;32(1):72-77.: 1/33 could help), since the most common dichotomy for GOS-E in most studies is between poor outcome [1–4: dead-vegetative-severe disability] versus good outcome [5–8: moderate disability-good recovery] (see: J Neurotrauma. 2019 Sep 1; 36(17): 2484–2492 and the references below.

Results/Discussion

-       Please justify the number of patients with good recovery in comparison with data from recent literature including pDOC. Previous long-term studies using GOS or GOS-E have described % of good recovery extremely low:

o   Steppacher I et al. Ann Clin Transl Neurol 2014: 1/59 (<2%) for UWS and 8/43 (19%) for MCS

o   Luauté J et al. Neurology 2010;75:246–252: 0/12 for UWS and 0/39 for MCS

o   Faugueras F et al. Brain Inj. 2018;32(1):72-77.: 1/33: 1/33 (3%)for UWS and 10/34 (29%) for MCS.

Reviewer 2 Report

Comments for Authors:

In the manuscript, the results are presented from a multi-year clinical study on patients of various ages and genders who suffered from advanced disorders of consciousness (pDoC) after brain injuries. A comparative analysis of neurological parameters was carried out in patients with favourable recovery and unfavourable outcome damaged consciousness.

The manuscript gives the impression of a carefully conducted research and analysis of the results.

The authors received new data, including on neurological indicators, such as sigma oscillations (sleepy winds), EEG reactivity, serum neuron-specific enolase, motor response, which can be useful for both clinicians and neuroscientists in the underdeveloped problem of diagnosis and rehabilitation of patients with severe cerebral disorders.

After correcting minor remarks, I recommend accepting the manuscript for publication.

Notes and Suggestions for Authors:

(1) Figure 2. The commentary to the Figure mentions the term VS, but Figure (2b) does not have this abbreviation. It is necessary to bring the Shape and the text to it into uniformity.

(2) Figure 3. In the commentary to the Figure, the abbreviations TBI, VL and ABI are mentioned, and their decoding is not in the caption to the figure, in the List of abbreviations and in the text. This should be corrected.

(3) The List of abbreviations must be complete.

(4)  The List of abbreviations.  For ease of reference, the abbreviations in the List are usually listed in alphabetical order.
